

# Reproductive biology of the biofuel plant *Jatropha curcas* in its center of origin

Manuel Rincón-Rabanales, Laura I. Vargas-López,
Lourdes Adriano-Anaya, Alfredo Vázquez-Ovando,
Miguel Salvador-Figueroa and Isidro Ovando-Medina

Instituto de Biociencias, Universidad Autónoma de Chiapas, Tapachula, Chiapas, Mexico

## ABSTRACT

In this work, we studied the main characteristics of flowering, reproductive system and diversity of pollinators for the biofuel plant *Jatropha curcas* (L.) in a site of tropical southeastern Mexico, within its center of origin. The plants were monoecious with inflorescences of unisexual flowers. The male flowers produced from 3062–5016 pollen grains (266–647 per anther). The plants produced fruits with both geitonogamy and xenogamy, although insect pollination significantly increased the number and quality of fruits. A high diversity of flower visiting insects (36 species) was found, of which nine were classified as efficient pollinators. The native stingless bees *Scaptotrigona mexicana* (Guérin-Meneville) and *Trigona* (*Tetragonisca*) *angustula* (Latreille) were the most frequent visitors and their presence coincided with the hours when the stigma was receptive. It is noteworthy that the female flowers open before the male flowers, favoring xenogamy, which may explain the high genetic variability reported in *J. curcas* for this region of the world.

## INTRODUCTION

*Jatropha curcas* (L.) (Euphorbiaceae), possibly native to Mexico and Central America (*Ovando-Medina et al., 2013*; *Salvador-Figueroa et al., 2015*), is considered the most promising non-edible plant for the production of biofuels. Many countries have established programs for its commercial cultivation (*Renner & Zelt, 2008*). In the Mesoamerican region, where the greatest genetic diversity of populations of *J. curcas* has been found (*Salvador-Figueroa et al., 2015*; *Basha et al., 2009*; *Ambrosi et al., 2010*; *Ovando-Medina et al., 2011*), monocultures are being established, for example in Guatemala and the Mexican states of Chiapas and Michoacan (*Ovando et al., 2009*).

When establishing new extensive crops, a multitude of factors must be taken into account, among which stands out reproductive biology (*Silva & Torezan, 2008*), i.e., knowledge about flowering, phenological behavior, sexual system, and fruit and seed production.

Studies on floral biology and pollination ecology of *J. curcas* have been conducted, mainly in regions where this species is exotic, as in India (*Sukarin, Yamada & Sakaguchi, 1987*; *Raju & Ezradanam, 2002*; *Bhattacharya, Datta & Kumar, 2005*; *Dhillon et al., 2006*; *Chang-Wei et al., 2007a*; *Quin et al., 2007*; *Rianti, Suryobroto & Atmowidi, 2010*; *Kaur,*

Corresponding author
Isidro Ovando-Medina,
isidro.ovando@unach.mx

*Dhillon & Gill, 2011*). Following from these studies, it is known that *J. curcas* is a monoecious species that presents geitonogamy and xenogamy; however, the first process, also called self-pollination, is prevalent. This could explain the low genetic diversity in Asian germplasm. It is possible that environment is affecting the prevalence of geitonogamy or xenogamy (*Heller, 1996*).

In the center of origin and diversity of *J. curcas* no such studies have been conducted, thus the sexual system and mode of pollination are unknown. Such knowledge could help design strategies to increase crop productivity; i.e. promoting efficient pollinators in commercial plantations. The aim of this study was to characterize the pollination process in *J. curcas* under field conditions at a site in southeastern tropical Mexico.

## MATERIALS AND METHODS

### Study site and biological material

The study was conducted within the living fences of a farm plot at Soconusco, Chiapas, Southern Mexico (14.5036 N, 92.1704 W, 58 m above sea level). In this zone the average annual temperature is 31 °C, the average annual humidity is 80%, and the average of rainfall is 2600 mm (Weather Station: 769043 MMTP of the Water National Commission, Mexico). The soil type was andosol with pH 5.7, 2.5% of organic matter and 0.2% of total nitrogen. Plants were selected based on their appearance (healthy and abundant foliage; approximately a 10% of the plants of the fence had foliar damage) and location (in a sunny area), on a 600 m transect. The living fence was 10 years old and underwent annual pruning. According to people living in the surrounding communities, the studied plants are considered toxic, with occasional cases of children intoxication caused by seed ingestion.

### Floral phenology

Flowering and fruiting dynamics were studied in 10 plants every 14 days during one year. We determined the average number of inflorescences per primary branch. In five inflorescences per plant, randomly selected, male and female flowers were counted. To estimate the time of flower anthesis and stigma receptivity, in other 10 plants, an inflorescence in the stage of flower buds was marked and observed daily (30 d) at intervals of every 10 min from 0700–1200 h. To estimate the production of pollen and ovules, inflorescences were collected from other 10 plants with closed flowers, and 20 Female Flowers (FF) and 20 Male Flowers (FM) were randomly selected. Pollen grains were extracted from FM and mounted in glycerinated gelatin on a slide. Pollen was quantified using a stereomicroscope and the average number of pollen grains was estimated per anther and per flower. In FF the number of carpels and the number of ovules per carpel was counted. With these data the pollen to ovule ratio was estimated.

In order to comprehend the reproductive process, five pollination treatments were established: 1) geitonogamy or artificial pollination with pollen from the same inflorescence (GEI), 2) Xenogamy (XEN) or artificial pollination with pollen from another plant, 3) Apomixis (APO), which was performed by removing the male

flowers and placing non-toxic white glue (Resistol®, Guadalajara, Mexico) on stigma, 4) Excluding Pollinators (ExP), and 5) Open Pollination (OpP). For each of the treatments 20 inflorescences were employed, one per plant (in total 20 plants for this experiment), which were covered, except in OpP, with tulle mesh bags of 1 mm mesh size. Fertilization was checked 14 days after pollination and the number of mature fruits per treatment was quantified at 55 days.

### Insect flower visitors and pollinators

To determine insect pollinators and visitors, observations were made on ten inflorescences from independent plants, from 0800–1700 h at intervals of 10 min. For this, the time of arrival at the flower was taken into account, the time they stayed, the resource used (nectar or pollen), and movement among flowers of the same inflorescence and between inflorescences of the same or another plant.

In addition insects visiting the flowers of *J. curcas* were collected on another living fence, with similar characteristics, but distanced 500 m from the study site to avoid interference with the previous experiments. For this part of the study, we sampled the area of influence of 100 plants. The insect collection was performed with entomological nets from 0600–1800 h. All insects captured were examined under stereoscope, dissected into head and thorax, and identified using (*Ayala, 1999*; *Michener, 2007*) taxonomic keys. Moreover, the pollen adhered to the legs of the insects was identified by microscopy.

The visiting insects were classified as: a) efficient pollinators, b) occasional pollinators, c) accidental pollinators, or d) pillagers. We used the following criteria: 1) number of individuals collected during different times of the day, 2) recurrence and time of visit to the male and female flowers, 3) behavior observed on the flowers, and 4) presence of *J. curcas* pollen (pure or mixed) on different parts of the body.

### Statistical analysis

To understand differences between visitor groups in the frequency and time of visit to male and female flowers, the Chi-square test was applied. The number of fruits, and the quality of these in the different treatments were compared by Analysis of Variance (ANOVA) and Tukey test ($\alpha = 0.05$).

## RESULTS

### Flowering and anthesis time

The plants studied had three flowering peaks (80%) in the months of April, May and September. In the period from December to February the plants showed no flowering. Subsequent to that, there were three periods of peak fruiting in the months of May, June and October. There were no fruits in the months from December to March.

It was found that, under the study conditions, *J. curcas* produces an average of 1.25 inflorescences per branch, and the number of female flowers per inflorescence was 1–11, with an average of 2.2. Meanwhile, the number of male flowers ranged from 35–198 flowers per inflorescence (mean: 106.7). It was found that the proportion of female/male flowers was 1:60. Pollen production by another ranged from 266–647 pollen grains

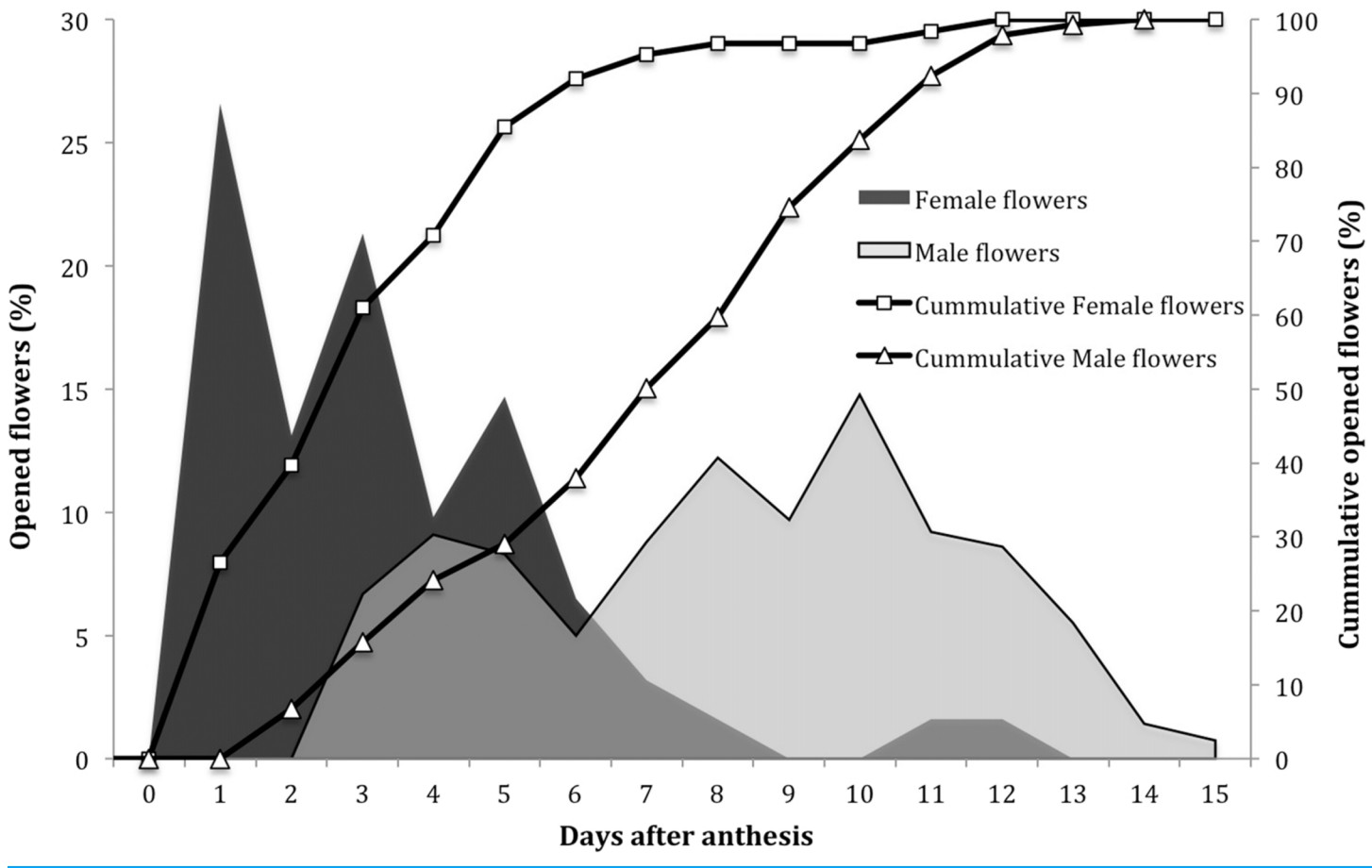

**Figure 1** Opening dynamics of female and male flowers in *Jatropha curcas* in the Soconusco region, Chiapas, Mexico.

(mean: 475.1) and per flower was 3062–5016 pollen grains (mean: 4224.4). The proportion of pollen grains per ovule was 1408:1.

The male and female flowers begin to open at 0800 h, presenting the maximum aperture of female flowers (64.29%) and male (55.75%) at 0900 h. The stigma was receptive at the time of 1000–1200 h.

## Pattern of flower opening

Flowers regularly opened over an average period of 15 days (Fig. 1). The pattern showed that the female flowers are the first to open and this process continues for eight days. Male flowers started opening two days after the female flowers, and lasted up to 13 days, with the highest peaks between days 8 and 10.

## Diversity of insect visitors

The variety of insect visitors to the flowers of *J. curcas* came from 36 species, which were grouped into four orders, 12 families and 16 genera as recognized (Table 1). Hymenoptera were the most diverse (75% of the species) and dominant (72.6% of the relative abundance) group, followed by Diptera (diversity of 19.4%; dominance of 26.3%).

**Table 1 Potential pollinators of *Jatropha curcas* in the region of Soconusco, Southern Mexico.**

| Order | Family | Genus | Species | Type of forage | Type of visitor | Relative abundance (%) |
|---|---|---|---|---|---|---|
| Hymenoptera | Apidae | *Apis* | *mellifera* Linneo | 1, 2 | OP | 1.1 |
| | | *Trigona* | *fulviventris* Guérin | 1, 2 | EP | 7.3 |
| | | *Trigona* | *fuscipennis* Friese | 1, 2 | EP | 1.1 |
| | | *Nannotrigona* | *perilampoides* Cresson | 1, 2 | OP | 0.4 |
| | | *Scaptotrigona* | *mexicana* Guérin-Meneville | 1, 2 | EP | 30.5 |
| | | *Tetragonisca* | *angustula* Lepeletier | 1, 2 | EP | 7.3 |
| | | *Oxitrigona* | *mediorufa* Cockerell | 1 | OP | 0.4 |
| | | *Melipona* | *beecheii* Bennett | 1, 2 | OP | 0.4 |
| | | *Melipona* | *solani* Cockerell | 1, 2 | OP | 0.4 |
| | | *Ceratina* | *capitosa* Smith | 1, 2 | OP | 0.4 |
| | | *Triepeolus* | sp. Robertson | 1 | PI | 0.4 |
| | Halictidae | *Agapostemon* | *nasutum* Smith | 1, 2 | EP | 7.3 |
| | | *Augochlora (Augochlora)* | *quiriguensis* Cockerell | 1, 2 | OP | 0.7 |
| | | *Augochlora (Oxystoglossella)* | *aurifera* Cockerell | 1, 2 | EP | 0.7 |
| | | *Augochlora (Augochlora)* | *smaragdina* Friese | 1, 2 | OP | 0.4 |
| | | *Halictus (Halictus)* | *ligatus* Say | 1, 2 | OP | 0.4 |
| | | *Halictus (Seladonia)* | *hesperus* Smith | 1, 2 | EP | 9.1 |
| | | *Lasioglossum (Dialictus)* | sp. 1 Robertson | 1, 2 | OP | 0.4 |
| | | *Lasioglosum (Dialictus)* | sp. 2 Robertson | 1, 2 | OP | 0.4 |
| | Formicidae | *Camponotus* | sp. 1 Mayr | 1 | OP, PI | 0.4 |
| | | *Crematogaster* | sp. 1 Lund | 1 | OP, PI | 0.4 |
| | | *Crematogaster* | sp. 2 Lund | 1 | OP, PI | 0.4 |
| | Sphecidae | – | sp. 1 | 1 | PI | 0.4 |
| | Sphecidae | – | sp. 2 | 1 | PI | 0.4 |
| | Sphecidae | – | sp. 3 | 1 | PI | 0.4 |
| | Vespidae | – | sp. 1 | 1 | AP, PI | 0.4 |
| | Vespidae | – | sp. 2 | 1 | AP, PI | 0.7 |
| Diptera | – | – | sp. 1 | 1 | PI | 0.4 |
| | Syrphidae | *Eristalis* | sp. 1 | 1 | EP | 7.3 |
| | Tachinidae | – | sp. 1 | 1 | EP | 17.0 |
| | Tachinidae | – | sp. 2 | 1 | AP, PI | 0.4 |
| | Syrphidae | – | sp. 1 | 1 | PI | 0.4 |
| | Bombyliidae | – | sp. 1 | 1 | PI | 0.4 |
| | Tephritidae | – | sp. 1 | 1 | PI | 0.4 |
| Coleoptera | Cerambycidae | – | sp. 1 | 1 | AP, PI | 0.7 |
| Hemiptera | Fulgoridae | – | sp. 1 | 1 | PI | 0.4 |

**Note:**
AP, accidental pollinator; EP, efficient pollinator; PI, pillager; OP, occasional pollinator. 1: nectar; 2: pollen.

We found three types of pollinators: a) accidental, including the fly Thachinidae sp. 2, Vespidae sp. 1, Vespidae sp. 2 and one species of Cerambicidae; b) occasional, comprising 14 species that included bees, ants, and wasps; and c) efficient, composed nine species,

**Table 2 Efficient pollinators of *Jatropha curcas* in the region of Soconusco, Southern Mexico.**

| Species | Individuals collected (n) | Type of pollen loads (number and percentage) | | |
|---|---|---|---|---|
| | | Pure loads | Mixed loads | Without loads |
| *Scaptotrigona mexicana* | 84 | 55 (65.5%) | 17 (20.2%) | 12 (14.3%) |
| *Tetragonisca angustula* | 19 | 14 (73.7%) | 1 (5.3%) | 4 (21.0%) |
| *Trigona fulviventris* | 19 | 14 (73.7%) | 2 (10.5%) | 3 (15.7%) |
| *Trigona fuscipennis* | 3 | 3 (100%) | – | – |
| *Halictus hesperus* | 25 | 15 (60%) | 3 (12%) | 7 (28%) |
| *Agapostemon nasutum* | 19 | 7 (36.9%) | 2 (10.5%) | 10 (52.6%) |
| Tachinidae sp. 1 | 49 | 31 (63.3%) | 1 (2.0%) | 17 (34.7%) |
| *Eristalis* sp. | 19 | 14 (73.7%) | 2 (5.3%) | 4 (21.0%) |
| *Apis mellifera* | 3 | 1 (33.3%) | 1 (33.3)% | 1 (33.3%) |

of which most were bees and two species of flies. The remaining insects were considered pillagers or nectar robbers (Table 1). Some of the individuals were carrying pure pollen loads of *J. curcas* in different parts of the body, while others carried *J. curcas* pollen mixed with other pollen types [*Ageratum* aff. *houstonianum* (Mill.), *Acacia* aff. *cornigera* (L.) Willd., *Tridax* aff. *procumbens* (L.), *Zea mays* (L.)]. Pollen from other species represented less than 10% of the total loads (Table 2).

## Frequency of visit

The activity of insects visiting the flowers of *J. curcas* was continuous, starting from 0600 h until shortly after 1800 h, showing a markedly bimodal behavior (Fig. 2). The main activity peaks coincided with increased secretion of female flower nectar during daylight 0700–0900 h. The greatest wealth of insects was recorded at 0900 and 1000 h (S = 8), registering the highest peak at 1400 h (S = 9) (Fig. 2).

We found differences in the frequency and time of insects visiting the female flowers ($\chi^2$ = 21.78, $p < 0.01$) and male ($\chi^2$ = 39.69, $p < 0.01$). Insects observed in the marked panicles were *S. mexicana*, *T.* (*Trigona*) *fulviventris* (Guérin), *T.* (*Trigona*) *fuscipennis* (Friese), *Agapostemon nasutus* (Smith), *Augochlora* (*Augochlora*) *quiriguensis* (Cockerell), *Augochlora aurifera* (Cockerell), *Augochlora* (*Augochlora*) *smaragdina* (Friese), *Halictus* (*Seledonia*) *hesperus* (Smith), Tachinidae sp. 1, *Eristalis* aff. (Williston), *Camponotus* (Mayr) and Vespidae sp. 1. Compared with other groups, the bees visited more flowers (43.2%) and stayed longer in resource foraging (38%). The second most important group was the Diptera with 39.3% (frequency) and 31% (time of visit). Specifically, bees visited a larger number of female flowers (55.6%) than male flowers (46.4%). Both bees and flies spent more time to visiting female flowers, while Vespidae sp. 1 had a preference for the male flowers. The Diptera foraged exclusively nectar, while bees collected nectar and pollen.

## *Jatropha curcas* reproductive system

We found differences in the production and quality of fruits and seeds from the different reproductive systems of *J. curcas* ($p < 0.001$). The highest percentage of fruit set was

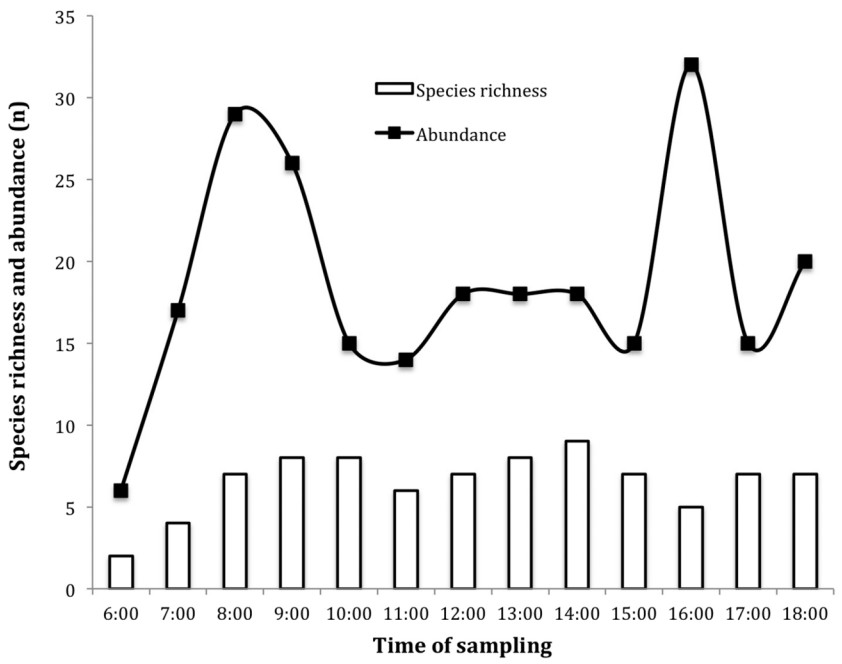

**Figure 2 Daily dynamics of insects visiting *Jatropha curcas* flowers in the Soconusco region, Southern Mexico.**

**Table 3 Comparison of characteristics of fruits and seeds obtained from different pollination treatments in *Jatropha curcas* in the Soconusco region, Chiapas, Mexico.**

|  | OpP | XEN | GEI | ExP | APO[*] | F[**] | p |
|---|---|---|---|---|---|---|---|
| Fruits per inflorescence (n) | 4.88[a] | 4.20[a] | 0.88[b] | 1.00[b] | 0.10[c] | 21.02 | 0.001 |
| Fruit diameter (cm) | 2.94[a] | 3.04[a] | 2.82[ab] | 2.64[b] | 2.9 | 2.98 | 0.035 |
| Fruit length (cm) | 3.29[a] | 3.26[ab] | 2.98[bc] | 2.85[c] | 3.1 | 4.73 | 0.004 |
| Fruit fresh weight (g) | 12.90[a] | 13.10[a] | 12.42[a] | 10.07[b] | 13.86 | 4.48 | 0.005 |
| Seeds (n) | 2.68[a] | 2.77[a] | 2.71[a] | 2.16[b] | 3.0 | 2.80 | 0.004 |
| Seed fresh weight (g) | 1.65[a] | 1.21[b] | 1.30[b] | 0.97[c] | 1.21 | 20.95 | 0.001 |

**Notes:**
OpP, open pollination; XEN, xenogamy; GEI, geitonogamy; ExP, excluding pollinators; APO, apomixis.
[*] Due to the reduced number of fruits, the apomixis treatment was not included in most of the ANOVA tests.
[**] One-way ANOVA and Tukey tests were performed. Different superscript letters in a row denote statistical differences among treatments, being "a" the highest value and "c" the lowest value.

recorded in free pollination treatments (OpP: 86.3 ± 2.2) and in xenogamy (XEN: 84.3 ± 6.3), being statistically equal between them; followed by the treatments with exclusion of pollinators (ExP: 18.1 ± 7.2) and geitonogamy (GEI: 16.2 ± 7.3). In the treatment of apomixis five fruits were formed, of which four were aborted and only one reached maturity (n = 55 female flowers). The only apomictic fruit had a high fresh weight, which contrasted with a low seed fresh weight, due to its thick endocarp.

Regarding the number of fruits that reached maturity, the OpP and XEN treatments were statistically superior to all other treatments ($p < 0.001$). The type of reproduction

also influenced the quality of fruits and seeds, as the longest fruits were recorded in the OpP and XEN treatments ($p < 0.001$) and heaviest seeds occurred in OpP treatment (Table 3).

## DISCUSSION

Some aspects of the floral biology of *J. curcas* in the Mexican tropics, such as the proportion of male and female flowers (1:60), differ from those reported in studies conducted with germplasm from other geographic areas. For example, *Raju & Ezradanam (2002)* reported in India that an inflorescence could produce 1–5 female flowers and 25–93 male flowers, with a ratio of male flowers to female flowers of 1:29. Also, *Pinto et al. (2009)* found 4–12 female flowers and 87–222 male flowers (relation: 1:20). This variable is not preserved and the differences depend on the genetic material, geographic region, climate, nutrition, time, and cultural practices, among other factors, which makes it a highly variable feature (*Bhattacharya, Datta & Kumar, 2005*; *Chang-Wei et al., 2007a*; *Pinto et al., 2009*). Pollen production per anther and per flower was higher than that reported in other studies, such as *Bhattacharya, Datta & Kumar (2005)*, who found that each flower produced 1617 pollen grains, with a ratio of pollen:ovule (P:O) of 539:1. In this regard, *Cruden (1997)* mentions that the P:O ratio is an indicator of the reproductive system. In the case of *J. curcas* the P:O relation is very high, which could favor the xenogamy. On the other hand, there could be a compensatory mechanism of pollen loss caused by the constant arrival of insect visitors to the inflorescences, i.e., reflecting a low efficiency in pollen transfer.

The anthesis of male flowers and female flowers under the conditions of this study occurred at 0700 h, which coincides with the findings of *Raju & Ezradanam (2002)*, who reported that the flowers open daily between 0530–0630 h. *Kaur, Dhillon & Gill (2011)* reported that the male flowers open between 0600 and 0700 h, while the female flowers open shortly after (0700–0800 h). The time it takes for the stigma to be receptive (1–2 h) is similar to that reported by *Bhattacharya, Datta & Kumar (2005)*. The opening pattern of flowers in the morning was related to attracting insects, because the availability of resources (nectar and pollen) is significantly higher in the morning.

The flowering stage of the plants studied occurred from March to November, which was consistent with that reported by *Sukarin, Yamada & Sakaguchi (1987)*, who recorded two flowering peaks, one in May and one in November. Instead, *Joker & Jepsen (2003)* observed that flowering occurs during the dry season with two flowering peaks, these authors mention that the plants bloom throughout the year in permanent wet conditions. While the fruiting stage occurred from April to December, contrary to that reported by *Toral et al. (2008)*, who recorded that the fruits are produced in winter when the plant loses its leaves.

The wealth of insects foraging on flowers of *J. curcas* located in the southeastern area of the Mexican tropics was high (Table 2). However, of the 32 species of potential pollinators, not all were efficient as pollinators of *J. curcas*, because not all visited the flowers of both sexes, or they did not transport pollen on their body, or they did not coincide with the period of stigma receptivity. According to these observations, it was

determined that an efficient pollinator for *J. curcas* was one that (i) visited several flowers of *J. curcas* during foraging (*Rianti, Suryobroto & Atmowidi, 2010*), (ii) frequently shifted from one flower to another (*Rianti, Suryobroto & Atmowidi, 2010*; *Free & Williams, 1977*), (iii) transported specific pollen abundantly on its body, and (iv) was observed sliding some part of its body on the receptive stigma.

The most abundant insects were Hymenoptera and Diptera, which is consistent with some previous studies in other areas (*Luo et al., 2011*), but contrasts in the case of Diptera that have been cited as efficient pollinators of flowers of *J. curcas*, but which are not always present or are less diverse and abundant (*Raju & Ezradanam, 2002*; *Bhattacharya, Datta & Kumar, 2005*). There is a high diversity and abundance of bees (19 species) and flies (9 species). Bees were generally small to medium size (5–10 mm) and particularly the native stingless bees were the most diverse, abundant, and with morphological characteristics correspondent to the floral syndromes of *J. curcas*. In general, the Diptera had body sizes 7–14 mm and abundant pilosity on the body, but their behavior does not allow cataloging them as efficient pollinators. Two exceptions were *Eristalis* sp. and Tachinidae sp. 1, which frequently visited male flowers looking for fresh nectar and transported pure pollen stuck onto their body for more than nine hours a day. They were observed simultaneously visiting female flowers for nectar during the period of maximum stigma receptivity facilitating xenogamy and geitonogamy, whereby they were classified as efficient pollinators. Our observations are consistent with studies by *Raju & Ezradanam (2002)*, who found that the Diptera *Chrysomya megacephala* (Fabricius) was an efficient pollinator that promoted xenogamy and geitonogamy. Conversely, *Rianti, Suryobroto & Atmowidi (2010)* reported only *Eristalis tenax* (L.) as an infrequent visitor and not an efficient pollinator for *J. curcas* in West Java.

Native stingless bees *S. mexicana*, *T.* (*T.*) *angustula*, *T.* (*T.*) *fulviventris*, *T.* (*T.*) *fuscipennis*, *H.* (*S.*) *hesperus* and *A. nasutus* are of small body size (5–8 mm), with special structures for transporting pollen and a great quantity of specialized hairs (*Roubik, 1989*). It is possible that these characteristics enable them to efficiently perform the flow of pollen to the stigma of *J. curcas*, as has been observed in other species of bees (*Raju & Ezradanam, 2002*; *Bhattacharya, Datta & Kumar, 2005*; *Rianti, Suryobroto & Atmowidi, 2010*; *Atmowidi, Rianti & Sutrisna, 2008*).

In the present study, *A. mellifera* (L.) was infrequent in relation to other efficient pollinators (n = 3) and recorded only between 0800 and 0900 h. Additionally, two individuals transported *J. curcas* pollen in great abundance, in the head, chest and legs, while one of them transported pollen from different plant species. This result is contrary to most previous studies where *A. mellifera* has been registered as the most efficient pollinator in *J. curcas* (*Raju & Ezradanam, 2002*; *Bhattacharya, Datta & Kumar, 2005*; *Dhillon et al., 2006*; *Quin et al., 2007*; *Rianti, Suryobroto & Atmowidi, 2010*; *Kaur, Dhillon & Gill, 2011*; *Chang-Wei et al., 2007b*). We classify this species as occasional pollinator for *J. curcas* at this site in the Mexican tropics, and it can perform both geitonogamy and xenogamy.

The polylectic behavior of stingless bees has been reported in previous work, and this is due to the different sources of pollen that they forage for food. However, it has been

reported that these bees can temporarily present an oligolectic strategy, taking advantage of a single source of food, as it occurs with *Cocos nucifera* (L.), *Manguifera indica* (L.), *Carica papaya* (L.), *Citrus limon* (L.) and *Capsicum annuun* (L.) (*Martínez et al., 1994*). Our data show that the efficient pollinating bees have oligolectic or monolectic behavior, at least during the season studied. Among those bees are included *S. mexicana* and *T.* (*T.*) *angustula*, which live in nests organized within tree trunks, have reduced foraging dispersion, and intensively use the floral resources available, for which they have a potential as inducers of pollination of extensive crops of *J. curcas*.

The increased production of fruits (86.3%) was recorded in open-pollinated plants and in xenogamy treatment (84.3%), consistent with the findings of other authors (*Raju & Ezradanam, 2002*; *Dhillon et al., 2006*; *Chang-Wei et al., 2007a*). However, a low fruit set (50–53%) is also reported in open-pollinated flowers (*Dhillon et al., 2006*). Regarding the phenomenon of apomixis, we found a very low rate (2.5%), similar to that found by *Santos, Machado & Lopes (2005)* in a semiarid region in Brazil (5%). In contrast, *Bhattacharya, Datta & Kumar (2005)* and *Kaur, Dhillon & Gill (2011)* reported that apomixis might be responsible for the formation of more than 30% of the fruits. *Chang-Wei et al. (2007a)* reported a moderate effect of apomixis (12%).

The genetic diversity of *J. curcas* in this Mesoamerican area and particularly in this region of Mexico is high (*Ovando-Medina et al., 2013*; *Basha et al., 2009*; *Ambrosi et al., 2010*; *Ovando-Medina et al., 2011*), and this may be largely due to the efficient pollination service by native stingless bees, to the strong trend for protogyny leading to pollination by xenogamy, and to the low prevalence of geitonogamy. Based in our results, strategies to improve the productivity of *J. curcas* in commercial plantations could be designed, for example increasing artificially the population of efficient pollinators as the stingless native bee *S. mexicana*.

## ACKNOWLEDGEMENTS

We thank the Hermanos Ávalos Ranch, in Tapachula, Chiapas, for allowing us to conduct the study in living fences on their property. We are thankful with El Colegio de la Frontera Sur, for the support to identify Halictideae and stingless bees.

### Funding

This work was partially financed by the project P/PROFOCIE-2014-07MSU0001H-11 (Secretaría de Educación Pública-México). The funders had no role in study design, data collection and analysis, decision to publish, or preparation of the manuscript.

### Grant Disclosures

The following grant information was disclosed by the authors:
Secretaría de Educación Pública-México: P/PROFOCIE-2014-07MSU0001H-11.

### Competing Interests

The authors declare that they have no competing interests.

## Author Contributions

- Manuel Rincón-Rabanales conceived and designed the experiments, performed the experiments, prepared figures and/or tables.
- Laura I. Vargas-López performed the experiments.
- Lourdes Adriano-Anaya analyzed the data, contributed reagents/materials/analysis tools.
- Alfredo Vázquez-Ovando performed the experiments, reviewed drafts of the paper.
- Miguel Salvador-Figueroa analyzed the data, contributed reagents/materials/analysis tools, wrote the paper, reviewed drafts of the paper.
- Isidro Ovando-Medina conceived and designed the experiments, wrote the paper, prepared figures and/or tables.

## Data Deposition

The research in this article did generate ordinary raw data.

## Supplemental Information

Supplemental information for this article can be found online at http://dx.doi.org/10.7717/peerj.1819#supplemental-information.

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
