# Peer review of "Reproductive biology of the biofuel plant Jatropha curcas in its center of origin"

_PeerJ, doi:10.7717/peerj.1819_

## Round 0.1 · original submission · Minor Revisions

Both reviewers find the manuscript acceptable and have only minor corrections. Please address the reviewers points, either through revision or in your rebuttal. I look forward to receiving your revised manuscript so we can move ahead with publication.

·

Basic reporting

mention if the plants of Jatropha are toxic or non toxic,
It would be appropriate to change the title and that is more specific, as are different agro-climatic conditions throughout southeastern Mexico of "Reproductive biology of the plant Jatropha curcas biofuel in Its center of origin" to "Reproductive biology of the plant Jatropha curcas biofuel from Soconusco, Chiapas. "describe in detail the climatic conditions of the Soconusco

Experimental design

Recommend that the study had been conducted in a plantation of a hectare, as live fences show the differences with a real plantation,
a soil analysis is recommended to determine the content of nitrogen, phosphorus and potassium correlate the results with flowering and weather conditions (temperature and average rainfall)

Validity of the findings

increase the number of plants evaluated because only were ten plants

Additional comments

1. mention if the plants of Jatropha are toxic or non toxic,
2. It would be appropriate to change the title and that is more specific, as are different agro-climatic conditions throughout southeastern Mexico of "Reproductive biology of the plant Jatropha curcas biofuel in Its center of origin" to "Reproductive biology of the plant Jatropha curcas biofuel from Soconusco, Chiapas. "
3. describe in detail the climatic conditions of the Soconusco,
4. increase the number of plants evaluated because only were ten plants
5. behavior is different than a plantation plants as a hedge, as well the proximity of plants live fence is one meter between plants

·

Basic reporting

I found this paper to be relatively straight forward and to the point. Being removed from botany for as long as I have did amount to constant breaks in the reading to refresh myself with certain terminologies and forgotten concepts. Once made comfortable I had no issue with the work. Aside from some minor grammatical suggestions I found it needed little editing.

Experimental design

No comments.

Validity of the findings

No comments.

---

## Round 0.2 · Minor Revisions

Both reviewers find the manuscript acceptable and have only minor corrections. Please address the reviewers points, either through revision or in your rebuttal. I look forward to receiving your revised manuscript so we can move ahead with publication.

---

## Round 0.3 · accepted · Accept

I really appreciate your prompt revision and responses to the reviewers' suggestions. I think the paper is ready to move forward, and I'm happy to accept it for publication.